# Dystonia: Sparse Synapses for D2 Receptors in Striatum of a DYT1 Knock-out Mouse Model

**DOI:** 10.3390/ijms21031073

**Published:** 2020-02-06

**Authors:** Vincenza D’Angelo, Emanuela Paldino, Silvia Cardarelli, Roberto Sorge, Francesca Romana Fusco, Stefano Biagioni, Nicola Biagio Mercuri, Mauro Giorgi, Giuseppe Sancesario

**Affiliations:** 1Department of Systems Medicine, Tor Vergata University of Rome, via Montpellier 1, 00133 Rome, Italy; dangelo@med.uniroma2.it (V.D.);; 2Santa Lucia Foundation, via del Fosso di Fiorano 64, 00143 Rome, Italy; 3Department of Biology and Biotechnology “Charles Darwin”, Sapienza University of Rome, Piazzale A. Moro 5, 00185 Rome, Italystefano.biagioni@uniroma1.it (S.B.)

**Keywords:** dystonia, striatum, D2 receptors, synapses, dopamine volume transmission

## Abstract

Dystonia pathophysiology has been partly linked to downregulation and dysfunction of dopamine D2 receptors in striatum. We aimed to investigate the possible morpho-structural correlates of D2 receptor downregulation in the striatum of a DYT1 Tor1a mouse model. Adult control Tor1a+/+ and mutant Tor1a+/− mice were used. The brains were perfused and free-floating sections of basal ganglia were incubated with polyclonal anti-D2 antibody, followed by secondary immune-fluorescent antibody. Confocal microscopy was used to detect immune-fluorescent signals. The same primary antibody was used to evaluate D2 receptor expression by western blot. The D2 receptor immune-fluorescence appeared circumscribed in small disks (~0.3–0.5 µm diameter), likely representing D2 synapse aggregates, densely distributed in the striatum of Tor1a+/+ mice. In the Tor1a+/− mice the D2 aggregates were significantly smaller (µm^2^ 2.4 ± SE 0.16, compared to µm^2^ 6.73 ± SE 3.41 in Tor1a+/+) and sparse, with ~30% less number per microscopic field, value correspondent to the amount of reduced D2 expression in western blotting analysis. In DYT1 mutant mice the sparse and small D2 synapses in the striatum may be insufficient to “gate” the amount of presynaptic dopamine release diffusing in peri-synaptic space, and this consequently may result in a timing and spatially larger nonselective sphere of influence of dopamine action.

## 1. Introduction

Dystonia is a disorder of movement characterized by disturbed agonist-antagonist muscle activation [1], with particular difficulty switching between sequential muscles involved in a complex task, either voluntary or involuntary [2]. Symptomatic dystonia can be observed in various neurological disorders, including cerebrovascular diseases, Parkinson’s disease, and Wilson’s disease [3]. Unlike symptomatic dystonia, no pathologic correlate is still known for many dystonic disorders categorized as idiopathic dystonia, and further divided into generalized, focal, and segmental dystonia [4]. 

In the past, various dystonic forms were often interpreted in psychological or psychiatric terms until the late 19th century [3,5], when the descriptions of familial forms of generalized primary torsion dystonia suggested that it is an organic disease [6,7]. The identification of genetic mutations in some families in the late 20th century established an organic framework for idiopathic dystonia (see 3).

The most common and severe form of genetic dystonia is caused by a 3 bp deletion (GAG) in the coding region of the TOR1A (DYT1) gene, which results in a defective protein called torsinA [8]. Although hereditary, DYT1 dystonia has a childhood or adolescent clinical onset [4]. However, significant neurodegeneration could not be documented at the histological level in any brain areas of patients with DYT1 dystonia [9,10], suggesting that dystonia pathophysiology is determined by functional rather than structural abnormalities. Several neurophysiological and neuroimaging studies, as well as new genetic insights, have been so far performed in DYT1 dystonia, contributing to define functional abnormalities in the basal ganglia [11,12,13,14,15]. Clinical neuroimaging studies have revealed decreased caudate-putamen dopamine D2 receptor availability in DYT1 patients compared to controls [16,17]. Moreover, reduced striatal D2 receptor binding and protein level have also been reported in several different DYT1 experimental models [18,19,20]. Aside from D2 receptor downregulation, multiple lines of evidence demonstrated reduced coupling between the D2 receptor and its cognate G proteins, as well as severely altered D2 receptor electrophysiological plasticity in the striatum but not in the *substantia nigra* [18,21,22,23,24,25]. Comparative studies on the functions of D1, D2, an A2A receptors, as well as of different neurotransmitters (dopamine, GABA, glutamate, acetylcholine) have been performed by Pisani et al. in mouse models of dystonia, demonstrating a selective downregulation and dysfunction of D2 receptors [18,21,23]. In addition, a recent paper has clarified the mechanisms of D2 receptor downregulation in the striatum, mediated by increased lysosomal degradation, associated in turn with lower levels of striatal RGS9-2 and spinophiling, opening a new approach to the therapy [26]. Therefore, it is generally assumed that abnormal striatal synaptic plasticity, and D2 receptor-dependent striatal outflow abnormalities have a leading role in determining basal ganglia pathophysiology in DYT1 dystonia [27,28]. The developmental profile of the aberrant D2 receptor function has been studied in DYT1 mutant mice, recording in cholinergic neurons an abnormal excitatory response to the D2 receptor agonist quinpirole since postnatal day 14, which persisted at three and nine months in hMT mice [22].

We aimed to investigate possible morpho-structural correlates of D2 receptor downregulation in striatum of an adult DYT1 knock-out mouse model. 

## 2. Results 

We first quantified the levels of D2 receptors on proteins extracted from the striatum. In line with a previous study [26] western blotting analysis revealed a significant ~ 30% reduction (*p* < 0.05) of D2 receptor levels in the striatum of mutant Tor1a+/− compared to control Tor1a+/+ mice (Figure 1).

Light microscopy immune-histochemistry demonstrated an intense D2 receptor brown peroxidase reaction product reactivity in the striatum (Figure 2A). In control Tor1a+/+, the striatum displayed D2 positive neuronal perikarya, peripherally outlined by an intense reaction product, and surrounded by a diffuse lighter neuropil staining. These data document a large distribution of D2 receptors on neuronal bodies, and neuropil of striatal neurons. In Tor1a+/− the D2 peroxidase reaction product appeared less intense around neuronal bodies, as well as in the neuropil of the striatum (Figure 2B), confirming the western blot analysis. However, the diffuse brown reaction product detectable by the D2 light microscopy immune-histochemistry can give just a rough idea of the D2 densitometric changes around neuronal bodies and neuropil, but does not allow a precise definition of the morpho-structural characteristics of the D2 receptor aggregates in control and mutant mice.

Immune-fluorescence images were acquired with a LSM700 Zeiss confocal laser scanning microscope (Zeiss, Germany): 5× and 20× objectives were used to define areas of interest in the dorsolateral striatum; distribution of D2 receptors was first acquired using 63× oil immersion lens (1.4 numerical aperture), and thereafter with an additional digital zoom factor (1×–1.5×–2×). Images of D2 immune-fluorescence acquired with a 63× oil immersion lens at first look appeared as a shiny galaxy in a starkly sky, with clusters of extremely small grains covering diffusely the neuronal compartments of the striatum, without apparent difference between perikarya and neuropil, whereas grains were rare and almost absent on the cell nuclei and in striatal axonal bundles (Figure 3). The density of D2 positive fluorescent grains was clearly different between the striatum of Tor1a+/+ and Tor1a+/− mice. In Tor1a+/+ the D2 positive grain were contiguous and even superimposed each other, whereas in the striatum of Tor1a+/− mice the D2 positive grains were close but separated from each other (Figure 3).

A better understanding of D2 receptors subcellular distribution came out in images acquired using a 63× oil immersion lens (1.4 numerical aperture) with an additional digital zoom factor (1×–1.5×–2×). The immune-fluorescent signal appeared extremely specific without background staining so that the D2 receptor localization appeared circumscribed in very small disks, roundish or elliptical in shape, of size variable with a diameter between ~0.3–0.5 µm in the striatum of Tor1a+/+ mice (Figure 4A).

In the striatum of Tor1a+/− mice, the D2 positive disks appeared sparse (Figure 4B), and the number and size of D2 positive disks and their total area per microscopic field was significantly lower in the striatum of Tor1a+/− than in corresponding areas of Tor1a+/+ mice (Figure 5A–C). Moreover, the mean distance between D2 positive disks was <1 µm for most of the disks in the striatum of Tor1a+/+ but > 2 µm for most of the disks in corresponding areas of Tor1a+/− mice (as can be easily seen in Figure 4).

However, quantitative analysis of the relative immune-fluorescence intensity per single D2 positive disk in the striatum was slightly less intense in Tor1a+/− than in Tor1a+/+ mice (Figure 6, Table 1), suggesting that the density of D2 receptors in large and small aggregates is substantially similar in control and mutant animals.

## 3. Discussion

The main finding of our work is that the reduced expression of D2 receptors in the Tor1a+/−striatum is associated with marked reduction in the number and size, but not in fluorescence intensity of the D2 positive residual aggregates. These results have however, some limitations in providing a morpho-structural correlate of the downregulation of D2 receptors in dystonia.

In our study, high-resolution confocal microscopy can give well-defined immune-fluorescent images of every D2 receptors’ aggregate in the microscopic field, but without configuring at the same time the mosaic of the surrounding cell structures, unlike electron microscopy immunocytochemistry. Indeed, immune-fluorescent D2 receptors appear arranged in homogeneous roundish shapes, variable in size with a diameter between ~0.3–0.5 µm, likely approaching dimensions of dopaminergic synapses on small dendritic spines (with shaft diameter ~0.04–1 μm and length ~0.1–2 μm) of striatal medium spiny neurons [29]. The D2 receptor positive disks likely represent the subcellular structures formed by D2 molecular aggregates, but we can just hypothesize the subcellular sites where the D2 aggregates are inserted in Tor1a+/+ and in Tor1a+/− mice.

Instead, at the electron microscopy level, in the striatum the D2 receptors were extensively localized to 38% of dendrites and 48% of spines of medium spiny neurons, and quite frequently on axon terminals [30]. Moreover, synapses formed by D2 immune-reactive terminals were not always easy to be identified in the striatum due to a lack of pronounced pre- or postsynaptic densities, suggesting a significant D2 extra-synaptic localization [30]. Indeed, in the striatum 60%–70% of dopaminergic axon terminals, supposed to be dopamine release sites, do not make synaptic junctional complex [31]. On the other hand, while photomicrographs obtained by transmission electron microscopy can allow detailed morphological characteristics of synapses, they can just give a rough estimate of synaptic sizes, shapes [32], and of spatial distribution frequency, due to the high magnification and limited electron microscopic fields.

By confocal microscopy in our study, we can obtain a definite spatial arrangement of the D2 signals with an estimate of sizes and shapes of the D2 receptor aggregates, and of their distribution density through a relatively large microscopic field. D2 receptor aggregates have generally a disk-shape and a size in the range of dimension of dendritic spines, and so D2 receptor aggregates have morphological similarity with synapses which can be categorized as macular synapses among the three main types of synaptic junctions (macular, perforated, and horseshoe-shaped synapses) [33]. Therefore, we would try to approximately interpret the reduction in number and size of the D2 aggregates in the striatum of Tor1a+/− mice as a global reduction of D2 synapses, but without differentiating between the synaptic and extra-synaptic sites. Reduction in synapses’ number and size is a well-known mechanism reducing efficiency of neuronal connectivity. However, the residual D2 synapses in Tor1a+/− mice display immune-fluorescence intensity and macular shape similar to those seen in Tor1a+/+ mice, suggesting that the synaptic arrangement and intensity of D2 receptors can be preserved in mutant animals.

A reduced topographic density of D2 synapses would be particularly relevant for dopamine transmission in the striatum of Tor1a+/− mice. Indeed, dopaminergic transmission is predominantly characterized as nonsynaptic or peri-synaptic volume transmission, since most dopaminergic receptors are extra-synaptic, i.e., rather distant from directly opposing dopaminergic terminal varicosities [29]. Once released by tyrosine hydroxylase (TH) positive terminals, dopamine is normally not limited in a synaptic cleft but diffuses far away from release sites, traveling through extracellular spaces usually along an effective radius of 7 μm in three dimensions, until dopamine molecules bind to synaptic and extra-synaptic D1 and D2 receptors or are cleared up by presynaptic dopamine transporters [29,34,35].

In Tor1a+/− mice, sparse D2 synapses in the striatum have to face a normal density of dopamine axon terminals, since the TH-positive axonal networks in mutant animals do not show morphologically obvious abnormalities in the striatum [36]. Moreover, there was no significant difference in striatal dopamine content between mutant and control mice, while its metabolite homovanillic acid (HVA) was higher suggesting that dopamine turnover was significantly increased in the mutant mice [36,37], according to evidence for increased dopamine turnover observed in the striatum of DYT1 patients [38]. Therefore, in DYT1 mutant mice the dopamine peri-synaptic landscape appears specifically changed, since D2 synapses are sparse in the striatum, but the sites of presynaptic dopamine release appear to be preserved [36,37]. In this scenario, dopamine molecules diffusing away from release sites can have a minor probability of binding in contiguous D2 synapses [39], so that they may have a large radius of diffusion, until they are bound by D1 and D2 receptors and/or are cleared up by presynaptic dopamine transporters. On the basis of the peri-synaptic dopamine volume efflux model, we hypothesize that in DYT1 mutant mice the sparse D2 synapses may be insufficient to “gate” a relatively abundant dopamine efflux, and this consequently may result in a timing and spatially larger nonselective sphere of influence of dopamine action.

The model of sparse D2 synapses can help interpret some aspects of DYT1 physiopathology in basal ganglia. At first glance, it seems inexplicable that the baseline and amphetamine-stimulated dopamine release detected by microdialysis is lower in mutant DYT1 mice compared with controls, while at the same time the caudate-putamen tissue dopamine levels were normal [36]. We can interpret the reduced dopamine release in mutant animals to be compensatory to D2 sparse synapses. Moreover, sparse D2 synapses can explain why DYT1 dystonia individuals do not respond to medications that alter dopamine transmission [36], since D2 synapses are sparse, there would be already a relative ceiling action of endogenous dopamine quantal release from dopamine preserved synaptic terminals. Finally, sparse D2 synapses can clearly differentiate the morpho-functional correlates of basal ganglia physiopathology in dystonia and Parkinson’s disease, the one characterized as post-synaptic pathology with D2 sparse synapses insufficient to gate dopamine efflux from preserved dopaminergic terminals, the other as a presynaptic pathology with loss of striatal dopamine release terminals but conserved D1 and D2 receptors in experimental parkinsonian monkey [40]. Moreover, in tissues from parkinsonian patients who died without receiving L-dopa in the last weeks of life, the binding of a radioactive ligand to Dl or D2 was elevated 20%–60% above the neurological control tissue [41]. Accordingly, dopamine receptor agonists are useful medications even regarded as first choice to delay the starting of L-dopa therapy in Parkinson’s disease [42], but ineffective in DYT1 dystonia [36].

Although the sparse D2 synapses may have a relevant role in DYT1 physio-pathology, it cannot be inferred that the observed D2 alteration is primary and specific, since the D2 receptors distribution in the striatum was not compared with other receptors in our study. In a previous work, we have demonstrated that in DYT1 mice carrying human mutant torsinA (hMT), the enkephalinergic neurons express a higher cellular content of the neuropeptide enkephalin and of PDE10A, a key enzyme in the catabolism of the second messenger nucleotides [43]. Therefore, we can infer that the downregulation of D2 receptors, predominantly expressed by the medium spiny enkephalinergic neurons, seems to be a primary process not secondary to reduced cellular capacity of protein synthesis in DYT1 dystonia.

However, it is not known whether the downregulation of the D2 receptors is selective and specific to the physiopathology of dystonia, or it is part of a larger process involving altered expressions of other receptor types, and in particular of the adenosine 2A (A2A) receptor subtype. It is worth noting that in the enkephalin positive striatal medium spiny neurons there is evidence for colocalization of the D2 receptors with the A2A receptors, which together form intramembrane cell complexes called heteromers, functionally interacting in a reciprocal antagonistic manner [44]. Likely, sparse D2 synapses may also result in an impaired D2/A2A interaction in DYT1 dystonia.

Further studies should investigate the synaptic and extra-synaptic distribution of the sparse D2 aggregates and of D2/A2A heteromers, evaluating their relationship with dopamine volume transmission in the physiopathology of basal ganglia microcircuits in DYT1 mutant mice. Moreover, clarifying the developmental morpho-structural characteristics of the D2 receptor aggregates and of D2/A2A etheromers in the DYT1 mouse model would be helpful to develop preventative treatment before clinical manifestation of disease.

## 4. Materials and Methods

### 4.1. Animals and Methods

C57BL/6 Tor1a+/− knock-out transgenic mice, that mimic the loss of function of the *DYT1* dystonia *TOR1A* mutation [45], were bred at the Santa Lucia Foundation Animal Facility; mice were sacrificed at 5–6-months-old. DNA was isolated and amplified from 1- to 2-mm tail fragments with the Extract-N-Amp Tissue polymerase chain reaction (PCR) kit (XNAT2 kit; Sigma-Aldrich, Milan, Italy, and genotyping performed as reported by Bonsi et al. [26]. All the efforts were made to minimize the number of animals utilized and their suffering. Treatment and handling of mice were carried out in compliance with both the EC and Italian guidelines (2010/63EU, D.lgs. 26/2014; 86/609/EEC; D.Lvo 116/1992), according to experimental protocols approved by the Animal Ethics Committee of the University of Rome “Tor Vergata” (D.M. 153/2001-A and 43/2002-A), and by the Santa Lucia Foundation Animal Care and Use Committee (D.M. 9/2006-A), and authorized by the Italian Ministry of Health (authorization # 223/2017-PR).

For biochemical studies, the animals (*n* = 5 Tor1a+/+, *n* = 5 Tor1a+/− knock-out) were killed by cervical dislocation, and their brains were removed rapidly and placed on an ice-cold plate. Thick brain sections were cut with an Oxford vibratome, and the caudate-putamen was dissected out rapidly from both hemispheres, and promptly frozen in liquid nitrogen and stored at –80 °C [46].

For morphological studies, the animals were deeply anesthetized with tiletamine/zolazepam (80 mg/Kg) and xylazine (10 mg/Kg), and perfused trans-cardially with 1% heparin in a 50 mL 0.1 M sodium phosphate buffer, and with 250 mL 4% paraformaldehyde in a 0.1 M sodium phosphate buffer (pH 7.4). The brains were removed immediately and post-fixed in the same fixative solution overnight at 4 °C. Coronal brain sections 40-μm-thick were cut with an Oxford vibratome across the entire rostro-caudal extent of the basal ganglia, and collected and stored at 4 °C in a 0.1 M phosphate buffer that contained 0.02% sodium azide, as previously described [43].

### 4.2. Quantitative Analysis of D2 Protein

The quantitative analysis of D2 receptor expressions in the striatum was assessed by western blotting. Tissues were lysated in 10 mM Tris–HCl pH 8, 150 mM NaCl, 1% *v*/*v* Triton X-100, 0.1% *w*/*v* sodium deoxycholate, 0.1% *w*/*v* sodium dodecyl sulfate, 1 mM EDTA, 5 mM β-mercaptoethanol, 1 mM PMSF, and protease inhibitor cocktail (Sigma-Aldrich). Thirty µg of proteins were loaded on a 9% SDS polyacrylamide gel and subjected to electrophoresis under reducing condition. The proteins were then transferred to a nitrocellulose membrane (Bio-Rad). The blots were incubated overnight at 4 °C with a rabbit polyclonal anti-D2 receptor antibody (1:1000, AB5084P; Millipore); or mouse anti-β-actin (1:10,000; Sigma-Aldrich) as a reference standard. Immune-reactive bands were revealed by horseradish peroxidase-conjugated secondary antibodies (1:10,000, Jackson Immunoresearch), incubated in a lumi-light-enhanced chemiluminescence substrate (Bio-Rad) and exposed to chemidoc (Bio-Rad). Densitometric analysis of scanned blots was performed using the NIH ImageJ version l.29 program (NIH, Bethesda, MD, USA).

### 4.3. Immune-Histochemistry

Coronal brain sections including the sensorimotor cerebral cortex, and the caudate-putamen were processed for identification of D2 positive receptors. Briefly, free floating sections were washed three times with Tris-buffered saline, pH 7.4, and endogenous peroxidase activity was inactivated by incubation for 5 min in Tris-buffered saline that contained 2% H2O2. Sections were rinsed with a Tris-buffered saline that contained 0.1% Triton X-100, and incubated with 2% normal goat serum, followed by overnight incubation at 4 °C with the primary polyclonal antibody (1:500) rabbit anti-D2 receptors (AB5084P, Millipore). The primary antibody was detected using a biotinylated secondary antibody (Vectastain ABC kit; Vector Laboratories, Burlingame, CA, USA), and an avidin horseradish peroxidase/diaminobenzidine/H2O2 chromogen system (Sigma-Aldrich, Milan, Italy). The specificity of the immunocytochemical reaction was confirmed by the absence of staining after omission of the primary antibody. After the diaminobenzidine reaction, the sections were rinsed with Tris-buffered saline and mounted on gelatine-coated slides, dehydrated, and cover-slipped with Permount for light microscopy examination. The sections were observed and photographed with a light microscope (Olympus BX51, Tokyo, Japan) equipped with an automatic micro camera (LeicaDC 300 F, Q550 IW Soft, Wetzlar, Germany).

### 4.4. Immune-Fluorescence Techniques and Confocal Microscopy

To evaluate the D2 receptor immune-fluorescence in striatal neurons, slices processing and confocal image acquisition were performed according to Bonsi et al. [26]. Mice were deeply anesthetized and perfused with cold 4% paraformaldehyde in a 0.12 M phosphate buffer (pH 7.4). The brain was post-fixed for at least 3 h at 4 °C and equilibrated with 30% sucrose overnight. Coronal brain sections (40 μm thick) were cut with a freezing microtome. Slices were dehydrated with serial alcohol dilutions (50%–70%–50%) and then incubated 1 h at room temperature in a 10% donkey serum solution in PBS 0.25%-Triton X-100 (PBS-Tx). The primary rabbit anti-D2 antibody (AB5084P, Millipore) was utilized (1:500, three days at 4 °C). The following secondary antibodies were used (1:200, room temperature, 2 h): Alexa 488 and Alexa 647 (Invitrogen), and cyanine 3 (cy3)-conjugated secondary antibodies (Jackson ImmunoResearch, Cambridge House, UK). Cell nuclei were detected with a blue-fluorescent DNA stain by 4′,6-diamidino-2-phenylindole (DAPI) (D9542, Sigma-Aldrich, Milan, Italy). After washout, slices were mounted on plus polarized glass slides with a Vectashield mounting medium (Super Frost Plus; Thermo Fisher Scientific) and coverslipped.

Images were acquired with a LSM800 Zeiss confocal laser scanning microscope (Zeiss, Germany), with a 5×, 20× objective, or 63× oil immersion lens (1.4 numerical aperture) with an additional digital zoom factor (1×–1.5×–2×) under no saturation conditions. Single-section images (1024 × 1024) or *z*-stack projections in the *z*-dimension (*z*-spacing, 1 μm) were collected. *Z*-stack images were acquired to analyze the whole neuronal soma, which spans multiple confocal planes. The confocal pinhole was kept at 1, the gain and the offset were adjusted to prevent saturation of the brightest signal and sequential scanning for each channel was performed. The confocal settings, including laser power, photomultiplier gain, and offset, were kept constant for each marker. For quantitative immune-fluorescence analysis, images were collected from at least 3–4 slices processed simultaneously from each striatum (*n* ≥ 4 mice/genotype) and exported for analysis with the ImageJ software (NIH). Software background subtraction was utilized to reduce noise.

The intensity of D2 fluorescence per D2 positive spots was evaluated by Java image processing and ImageJ. Briefly, the D2 positive spots in immune-fluorescence microphotographs were randomly selected using a drawing selection circle closely around, and the red color image of D2 fluorescence intensity was analyzed by converting each pixel in greyscale (0–255). The minimum and maximum grey values within the selection were detected, and the mean grey value (the sum of the grey values of all the pixels in the selection area divided by the number of pixels) per spot was calculated.

### 4.5. Statistical Analysis

Data analysis was performed using the Statistical Package for the Social Sciences Windows, version 15.0 (SPSS). Descriptive statistics consisted of the mean ± SE for parameters with Gaussian distributions (after confirmation with histograms and the Kolgomorov–Smirnov test). The homogeneity of the variance was evaluated by Levene’s test. Comparison among groups was performed with the ANOVA one-way. *P*-value < 0.05 was considered statistically significant.

## 5. Conclusions

The reduced expression of D2 receptors in the striatum of Tor1a+/− dystonic mice is associated with marked reduction in number and size of the D2 positive synapses. D2 sparse and small synapses may be insufficient to “gate” a relatively abundant dopamine release diffusing extrasynaptically through the interstitial space among striatal neurons. This may result in a longer duration and larger sphere of influence of dopamine transmission, making its action non-selective in striatal micro-circuitries. Our study suggests a relevant role for dopamine volume transmission in DYT1 dystonia physiopathology.

## Figures and Tables

**Figure 1 ijms-21-01073-f001:**
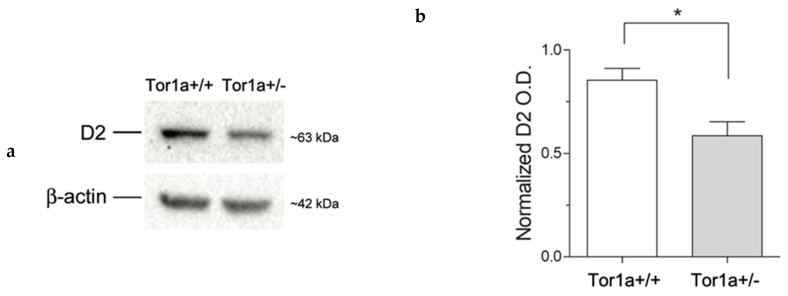
(**a**) Comparative immunoblots of D2 receptors from the striatum of control Tor1a+/+, and mutant Tor1a+/− mice. β-actin content was detected as internal reference standard. (**b**) Densitometric analysis of relative optical density (OD) on D2 receptors immune-stained bands. Results were expressed as the mean ± SEM of the values obtained for each separate hemisphere from four mice per group. * *p* < 0.05.

**Figure 2 ijms-21-01073-f002:**
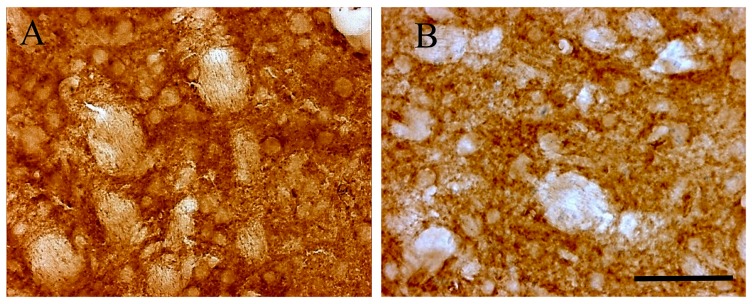
Representative microphotographs of D2 receptor immune-histochemistry in control Tor1a+/+ (**A**), and mutant Tor1a+/− knock-out (**B**) mice. The brown reaction product is clustered around neuronal bodies and diffused in the neuropil. Scale bar in **B** = 100 μm.

**Figure 3 ijms-21-01073-f003:**
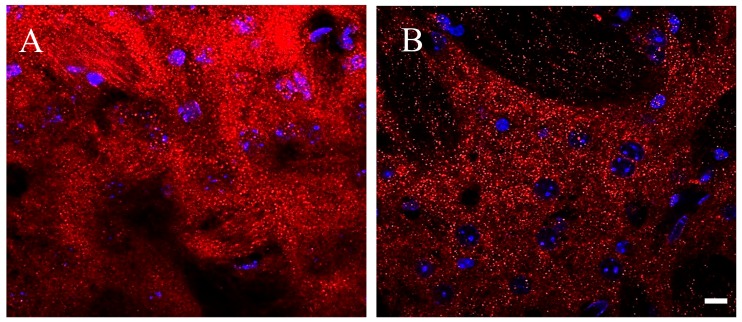
Representative immune-fluorescence microphotographs of confocal laser scanning microscopy (63×), double-labeled for D2 receptors and for nuclei in the striatum of control Tor1a+/+ (**A**), and of mutant Tor1a+/− knock-out (**B**) mice. D2 receptor immune-labeling is visualized in red-Cy3 fluorescence, while nuclei are visualized by DAPI fluorescence in blue. Bar in **B** = 10 µm.

**Figure 4 ijms-21-01073-f004:**
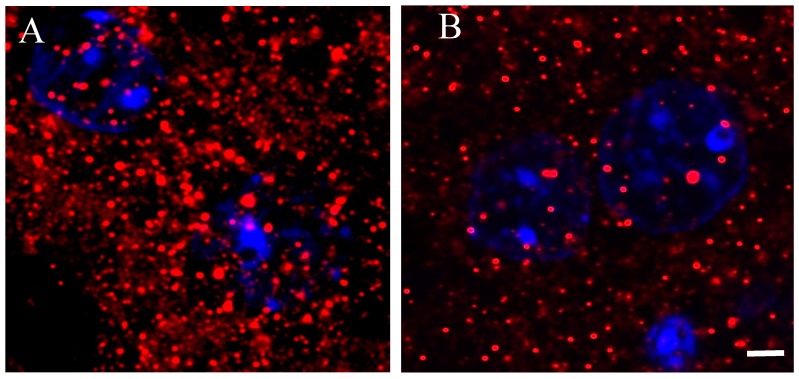
Representative immune-fluorescence microphotographs of high magnification (63× plus zoom factor 1×–1.5×–2×) confocal laser scanning microscopy, double-labeled for D2 receptors and for nuclei in the striatum of control Tor1a+/+ (**A**), and of mutant Tor1a+/− knock-out (**B**) mice. D2 receptor immune-labeling is visualized in red-Cy3 fluorescence, while nuclei are visualized by DAPI fluorescence in blue. Bar in **B** = 2 µm.

**Figure 5 ijms-21-01073-f005:**
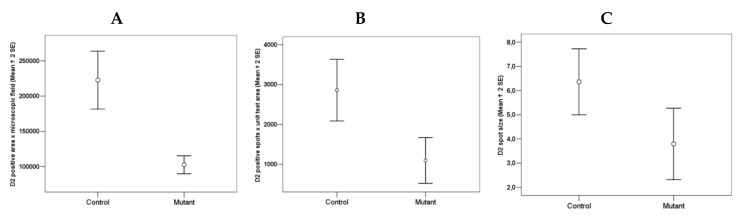
Comparative densitometric analysis of the D2 immune-fluorescence in the striatum of control Tor1a+/+ and mutant Tor1a+/− knock-out transgenic mice. (**A**) Total D2 positive area per microscopic field through 63× oil immersion objective. df = 1, F = 31.202; *p* < 0.0001. (**B**) Number of D2 positive disks per microscopic field through 63× oil immersion objective (1.4 numerical aperture) with an additional digital zoom factor (1×–1.5×–2×). df = 1, F = 13.406; *p* < 0.001. (**C**) Size of D2 disk (μm^2^) through 63× oil immersion objective (1.4 numerical aperture) with an additional digital zoom factor (1×–1.5×–2×). df = 1, F = 6.543; *p* < 0.04. Results were expressed as the mean + SE of average values detected within the randomly selected microscopic areas of the dorsolateral caudate-putamen of both hemispheres from four mice in each group. Comparison among groups was performed with the ANOVA one-way.

**Figure 6 ijms-21-01073-f006:**
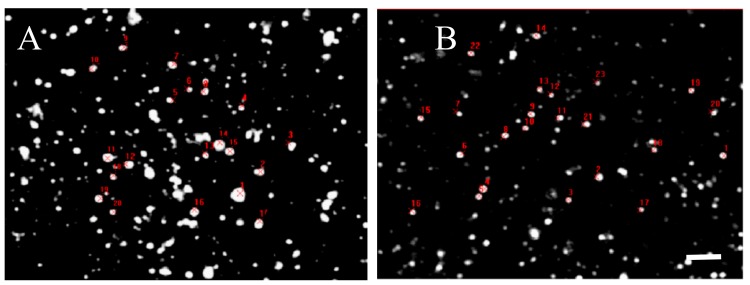
Table 1—Comparative fluorescence intensity of D2 receptor spots evaluated by Java image processing and ImageJ in microphotographs from confocal laser scanning microscopy. D2 positive spots were randomly selected and marked in red in control Tor1a+/+ (**A**), and mutant Tor1a+/− knock-out (**B**) mice. Bar in **B** = 2 µm. Table 1—Results were expressed as the mean + SE of the area and the average grey values (0–255) within the selected spots of the dorsolateral caudate-putamen of both hemispheres from three mice in each group. Marked decrease of the area (see analysis in Figure 5), but not of the grey level in D2 positive spots was observed in mutant Tor1a+/− knock-out mice, compared with controls.

**Table 1 ijms-21-01073-t001:** Control Mutant.

**Spots**	**Area µm^2^**	**Grey level**	**Area µm^2^**	**Grey Level**
**Mean**	**6.73**	**252.00**	**2.4**	**216.18**
± SE	3.41	0.30	0.16	13.63

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
