# Peer review of "Dystonia: Sparse Synapses for D2 Receptors in Striatum of a DYT1 Knock-out Mouse Model"

_ijms, 2020, doi:10.3390/ijms21031073_

Round 1
Reviewer 1 Report
In this manuscript entitled; Dystonia: Sparse Synapses for D2 receptors in Striatum of a DYT1 knock-out mouse model, the authors investigated the morpho-structural correlates of D2 receptor downregulation in striatum of DYT1 mouse model. This is an interesting study provided some details about D2 receptors aggregate in striatum of Tot1a+/- mice. The manuscript is well written with a good standard English. The manuscript is worthy of consideration for publication though a few questions need to be addressed first.
Have the authors tried to check D2 receptors downregulation in human brain sections from patients with Parkinson’s disease and compare it with the current results from the mice model? In page 2, line 64, I think it is important to show the western blot data and include it in figure 1. Typo errors: Page 1, remove the extra space between line 18 and 19. Page 2, line 56, please correct the font size. Page 2, line 70, immune-hystochemsitry should be immune-histochemistry. Page 5, line 117, correct the font for Tor1a+/+ and mutant Tor1a+/- . Page 5, table 1 is overlapped with figure 5, please reorganise it. Page 6, please remove the extra space between line 177 and 178. Page 7, line 210, 213 and 216, replace : with comma. Page 7, line 239-247, please correct the font size.Author Response
Please see the attachement

Reviewer 2 Report
The paper extends research from the same group on DR2D altered status in Dyt1 mice by focussing on the distribution of the synaptic buds in the striatum and providing some insight into the pathophysiology of Tor1a associated dystonia. The issue is relevant, the methods are sound but should be better defined and the conclusions are consequential.
There are some issues that need to be addressed before considering the work for publication.
1) the age of the mice at the time of analysis should be specified (DYT1 shows an age related variably expressed phenotype, which likely might be reflected in the underlying anatomo-physiological anomaly)
2) the quali/quantitative evaluation of the D2 receptors in the striatum is not "normalized" against the distribution of other striatal receptors. In such way it cannot be inferred that the observed alteration is primary and specific
3) again, considering the evolutive nature of the clinical manifestation in DYT1 patients, some consideration/observation on the longitudinal behaviour of the D2 receptors in the experimental model would be of great interest.
4) the table 1 (figure5) is unreadable being completely out of format
Minor point concerns the English syntax (see line 55, 59 and 159 for the inappropriate use of "eventual", 161-162 which appear obscure
Author Response
Please see the attachement
